

# Joint-level energetics differentiate isoinertial from speed-power resistance training—a Bayesian analysis

Bernard X.W. Liew[1,2], Christopher C. Drovandi[3,4], Samuel Clifford[3,4], Justin W.L. Keogh[5,6,7], Susan Morris[2] and Kevin Netto[2]

[1] School of Sports, Exercise, Rehabilitation Sciences, University of Birmingham, Birmingham, West Midlands, United Kingdom
[2] School of Physiotherapy and Exercise Sciences, Curtin University of Technology, Bentley, WA—Western Australia, Australia
[3] School of Mathematical Sciences, Queensland University of Technology, Brisbane, Queensland, Australia
[4] ARC Centre of Excellence for Mathematical and Statistical Frontiers, Brisbane, Queensland, Australia
[5] Faculty of Health Sciences and Medicine, Bond University, Queensland, Australia
[6] Sports Performance Research Centre New Zealand, Auckland University of Technology, Auckland, New Zealand
[7] Cluster for Health Improvement, Faculty of Science, Health, Education and Engineering, University of the Sunshine Coast, Queensland, Australia

Corresponding author
Bernard X.W. Liew,
LiewB@adf.bham.ac.uk,
LiewB@bham.ac.uk,
liew_xwb@hotmail.com

## ABSTRACT

**Background.** There is convincing evidence for the benefits of resistance training on vertical jump improvements, but little evidence to guide optimal training prescription. The inability to detect small between modality effects may partially reflect the use of ANOVA statistics. This study represents the results of a sub-study from a larger project investigating the effects of two resistance training methods on load carriage running energetics. Bayesian statistics were used to compare the effectiveness of isoinertial resistance against speed-power training to change countermovement jump (CMJ) and squat jump (SJ) height, and joint energetics.

**Methods.** Active adults were randomly allocated to either a six-week isoinertial ($n = 16$; calf raises, leg press, and lunge), or a speed-power training program ($n = 14$; countermovement jumps, hopping, with hip flexor training to target pre-swing running energetics). Primary outcome variables included jump height and joint power. Bayesian mixed modelling and Functional Data Analysis were used, where significance was determined by a non-zero crossing of the 95% Bayesian Credible Interval (CrI).

**Results.** The gain in CMJ height after isoinertial training was 1.95 cm (95% CrI [0.85–3.04] cm) greater than the gain after speed-power training, but the gain in SJ height was similar between groups. In the CMJ, isoinertial training produced a larger increase in power absorption at the hip by a mean 0.018% (equivalent to 35 W) (95% CrI [0.007–0.03]), knee by 0.014% (equivalent to 27 W) (95% CrI [0.006–0.02]) and foot by 0.011% (equivalent to 21 W) (95% CrI [0.005–0.02]) compared to speed-power training.

**Discussion.** Short-term isoinertial training improved CMJ height more than speed-power training. The principle adaptive difference between training modalities was at the level of hip, knee and foot power absorption.

## INTRODUCTION

Vertical jump performance and mechanics are associated with physical performances across a range of running-related sports, even at the recreational level (*Gonzalez-Rave et al., 2011*; *Hebert-Losier, Jensen & Holmberg, 2014*; *Rousanoglou et al., 2016*). To date, the squat (SJ) and countermovement jumps (CMJ) are the most commonly assessed vertical jump types (*Claudino et al., 2017*; *De Villarreal et al., 2009*), and as such are often used as markers of training-related improvements in sporting performance (*Taipale et al., 2014*; *Taipale et al., 2013*).

Resistance training for vertical jump improvements have typically focused either on augmenting muscular force, or muscular speed-power variables (*Cormie, McGuigan & Newton, 2011*). Resistance training focused on increasing muscular force capacity uses heavy external load magnitudes (e.g., >80% one repetition maximum [1RM]) (*Cormie, McGuigan & Newton, 2011*). This contrasts with training focused on increasing muscular speed-power, which incorporates lighter load magnitudes and quicker movement velocities (*Cormie, McGuigan & Newton, 2011*). There is convincing evidence from systematic reviews that either a force-focused or speed-power focused resistance training can similarly improve vertical jump performance relative to no training (*Markovic, 2007*; *Perez-Gomez & Calbet, 2013*). In other words, there is convincing evidence for a variety of resistance training modalities improving vertical jump in a variety of populations.

Despite the convincing evidence, most performance training studies comparing muscular force to muscular speed–power focused interventions have failed to identify significant between training modality differences in improvements (*Cormie, McGuigan & Newton, 2010a*; *De Villarreal, Izquierdo & Gonzalez-Badillo, 2011*; *Wilson et al., 1993*). One reason for this is that the between-group changes are much smaller than the within-group improvements, and so may be missed by traditional ANOVA-based Frequentist statistics (*Cormie, McGuigan & Newton, 2010a*; *De Villarreal, Izquierdo & Gonzalez-Badillo, 2011*; *Wilson et al., 1993*). Only one study has previously reported significantly greater vertical jump height gains with speed-power than force-focused training, where between-group effects could have been magnified by the use of percentage improvements from baseline (*Newton, Kraemer & Hakkinen, 1999*). To avoid missing a small but potentially beneficial intervention effect, recent studies have used descriptive methods based on binned effect sizes (*Cormie, McGuigan & Newton, 2010a*; *Jimenez-Reyes et al., 2016*), initially described by *Barker & Schofield (2008)* as the qualitative magnitude-based method. While this approach overcomes many of the limitations of the ANOVA-based approaches, a shortcoming of this approach is that the probabilities associated with the effect sizes are not quantified. Prescriptive decision making often hinges on weighing the probabilities against the magnitude of effect. In contrast, Bayesian statistics provide a framework where all

plausible between-group effect sizes can be easily interpreted using probabilities (*Mengersen et al., 2016*).

A second reason for the lack of consensus in identifying optimal training methods, may be the lack of mechanisms-based intervention studies evaluating joint-level energetics. Training studies on vertical jumps have focused on analyzing the linear force-velocity characteristics of the ground reaction force (GRF) (*Cormie, McGuigan & Newton, 2010a*; *Cormie, McGuigan & Newton, 2010b*). GRF analysis reveals only the limb-level force output which emerges from the net interaction of joint torques (*Bobbert, 2012*). In contrast, a smaller number of studies have identified inter-joint and inter-limb coordination as important variables which discriminates optimal and suboptimal jumping performance (*Bobbert & Van Soest, 2001*; *Yoshioka et al., 2010*). Common exercises used in speed-power training, such as hopping and jumping (*Lloyd et al., 2012*), exploit inter-joint and inter-limb coordination patterns (*Bobbert & Van Soest, 2001*; *Yen, Auyang & Chang, 2009*; *Yoshioka et al., 2010*), that may produce greater transfer to sporting performance than traditional high-force isoinertial training.

To date, it is unknown which, if any, resistance training method is superior in improving jump performance, and the joint-level mechanism(s) of effect. The primary aim of this study was to use Bayesian inference to compare the between-group differences in change of a force-focused resistance training (termed "isoinertial training") against a speed-power-focused training (termed simply as "speed-power training") on CMJ and SJ height, and the associated joint-level energetics. Based on the principle of exercise specificity, it was hypothesized that greater increases to CMJ and SJ heights and their joint-level power magnitude would occur in speed-power compared to isoinertial training.

## MATERIALS AND METHODS

### Study design, participants, randomization, blinding

The data presented in this manuscript represents the results of a sub-study from a larger project investigating the effects of two resistance training methods on load carriage running energetics. A full description of the overall study protocol has been published (*Liew et al., 2016*). We adhered to the CONSORT guidelines for the reporting of this trial (Fig. 1).

In brief, recreational novice runners (i.e., weekly running duration >45 mins) were recruited for the study. Exclusion criteria included medical conditions which prevented the safe performance of strenuous physical activities, running injuries in the past 12 weeks, surgeries within the past 12 months, and being pregnant. All participants had to provide written informed consent prior to study enrolment. This study was approved by the Curtin University's Human Research Ethics Committee (RD-41-14). The descriptive characteristics of the participants are detailed in Table 1.

A computer-generated sequence of random numbers was generated and allocation to training groups was concealed via sealed-opaque envelopes (*Liew et al., 2016*). Only the outcome assessor was blinded to the group allocation.

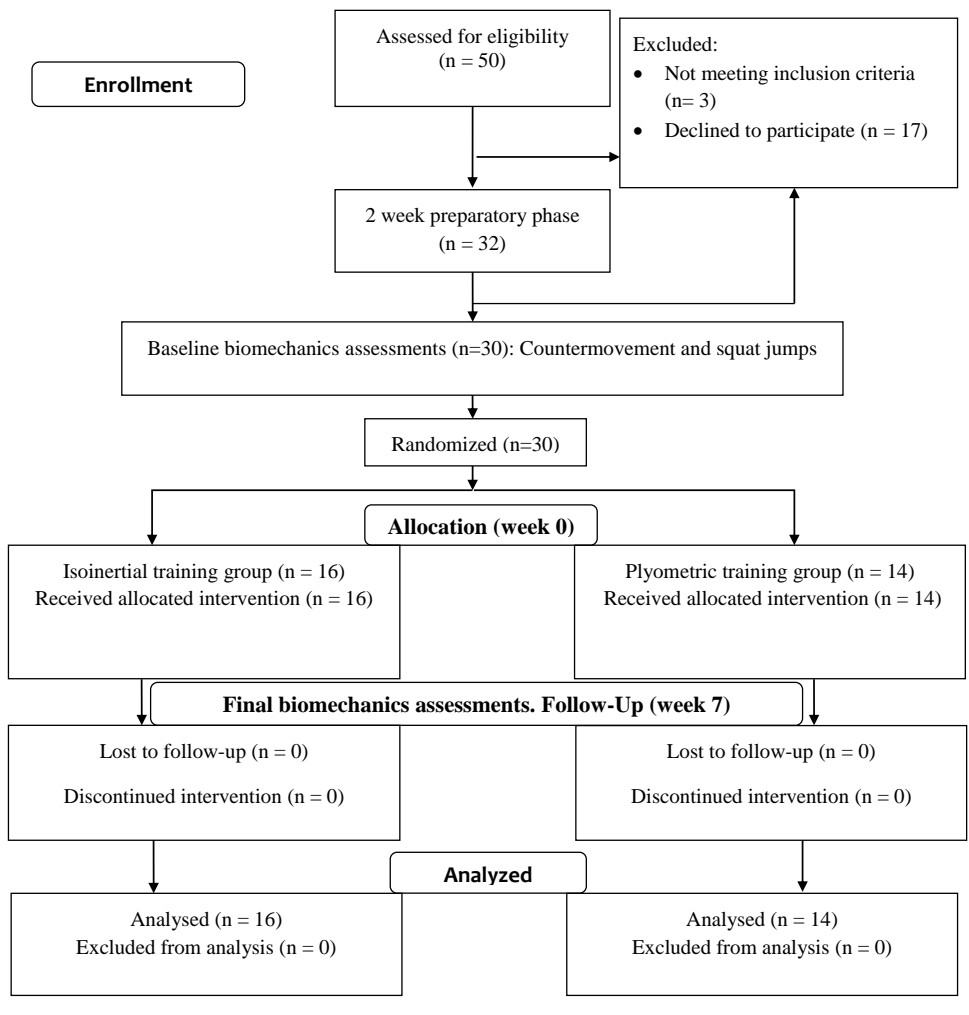

**Figure 1** **CONSORT Flow diagram.**

## Sample size

This study was originally statistically powered to detect a significant change in running leg stiffness for the primary study on load carriage running (*Liew et al., 2016*). With a standard deviation of 3,600 N/m and a correlation between repeated measures of 0.80, 24 participants were needed to detect a between-group difference of 3,000 N/m (*Dupeyron et al., 2013*), at a power of 0.80 ($\alpha = 0.05$). Thirty participants were recruited to allow for a 20% drop out proportion. For the present study, this provided a power of 0.80 to detect a between group difference in change effect size of 0.6 (i.e., between group difference in change of 2.6 cm, average standard deviation of 4.35 cm (*Newton, Kraemer & Hakkinen, 1999*)).

## Intervention
### Background

The two training programs were developed to improve load carriage running energetics (*Liew et al., 2016*). Details of the training protocol can be found in Table 2. One group

**Table 1  Participant's baseline characteristics.**

|  | Isoinertial ($n = 16$) | Speed-power ($n = 14$) | $p$ value |
|---|---|---|---|
| Age (years) | 30.8 (8.8) | 29.4 (9.9) | 0.691 |
| Body mass (kg) | 67.8 (14.0) | 69.2 (10.3) | 0.750 |
| Height (cm) | 172.4 (8.8) | 171.7 (6.4) | 0.811 |
| Gender (Male:Female) | 8:8 | 7:7 | 1.00 |
| Resistance training frequency over past 6 weeks (times/week) | 1.0 (0.8) | 1.7 (1.6) | 0.146 |
| Resistance training frequency over past 12 months (times/week) | 1.2 (1.2) | 3.0 (3.3) | 0.067 |
| Resistance training experience (B:I:A) | 12:4:0 | 6:5:3 | 0.085 |
| Running experience (years) | 9.6 (5.9) | 9.3 (6.3) | 0.903 |
| Running frequency over past 6 weeks (times/week) | 2.4 (1.3) | 3.0 (1.5) | 0.235 |
| Running distance over past 6 weeks (km/week) | 12.0 (13.7) | 21.8 (20.2) | 0.138 |
| Training attendance (maximum of 18) | 16.7 (1.8) | 16.9 (1.8) | 0.801 |

**Notes.**

Abbreviations: B, beginner; I, intermediate; A, advanced; Beginner, currently not resistance training or started but ≤2 months with a frequency of ≤1 session/week; Intermediate, currently doing resistance training and started within the last 2 to 6 months with a frequency of 2 to 3 sessions/week; Advanced, currently doing resistance training with ≥1 years' experience with a frequency of ≥4 sessions/week.

performed heavy resistance isoinertial training of the lower limb (*Liew et al., 2016*). Exercises in the isoinertial group focused on increasing muscular force, which included unilateral seated calf raises, lunges, and bilateral leg press. This training has been broadly used to augment military load carriage physical performance (*Knapik et al., 2012*).

The speed-power training comparative group, performed resistance training targeting known biomechanical requirements of load carriage running. For example, load carriage in running has been shown to require increased leg stiffness, knee power generation, and hip power absorption to maintain constant velocity (*Liew, Morris & Netto, 2016b*; *Silder, Besier & Delp, 2015*). Two of the three exercises in this group focused on increasing muscular speed-power capacity using externally loaded countermovement jumps (CMJ) and single leg (SL) hopping. A third exercise, isoinertial hip flexor cable pull, focused on augmenting load carriage running pre-swing energetics, and would not have a significant effect on jumping performance (*Deane et al., 2005*). The number of foot-contact repetitions used in the CMJ (345 contacts) and SL hopping (2,236 contacts) exercises across the training program was similar to previous speed-power training studies which were successful at improving vertical jump performance and reactive strength capacity (*Lloyd et al., 2012*; *Wilson, Murphy & Giorgi, 1996*).

The total training volume performed by each group differed due to the emphasis on ecological realism in training implementation. Using a simple metric of training volume (total repetitions across six weeks), the speed-power group performed a total of 2,581 repetitions (not accounting for hip flexor training), whilst the isoinertial group performed a total of 852 repetitions. Such experimental training design has precedence in previous training studies for sports (*De Villarreal, Izquierdo & Gonzalez-Badillo, 2011*; *Sáez*

**Table 2 Training protocol.** Averaged fortnightly training prescription per exercise session (see study protocol of absolute training volume).

| Weeks | Familiarization 1–2 | Training 3–4 | 5–6 | 7–8 |
|---|---|---|---|---|
| *Pre-randomization (common to all participants)—4 sessions total* | | | | |
| SL hopping (BW) | 2 sets × 20 hops | • Familiarization period used for determining starting 10RM load for hip flexor pull, leg press, calf raise, lunge <br> • 10RM load used to estimate 1RM load for the four exercises | | |
| CMJ | 3 sets × 3 reps | | | |
| Hip flexor pull | 1–2 sets × 10 reps × 10–15RM | | | |
| Leg press | 2–3 sets × 10 reps × 10–15RM | | | |
| Calf raise | 1–2 sets × 10 reps × 10–15RM | | | |
| Lunge | 1–2 sets × 10 reps × 10–15RM | | | |
| *Isoinertial group (3 sessions per week) —18 sessions total* | | | | |
| Leg press | | 2–3 sets × reps × 10 RM | 2–4 sets × 6 reps × 8RM | 2–4 sets × 4 reps × 6 RM |
| Calf raise | Time based criterion for load increment of weekly adjusted 1RM | 2–3 sets × reps × 10 RM | 2–3 sets × 6 reps × 8RM | 2–4 sets × 4 reps × 6 RM |
| Lunge | | 2–3 sets × reps × 10 RM | 2–3 sets × 6 reps × 8RM | 2–4 sets × 4 reps × 6 RM |
| *Speed-power group (3 sessions per week)—18 sessions total* | | | | |
| SL hopping | SL hopping and CMJ time based criterion for load increment. | 2–4 sets × 20 s × 2.2 Hz × 110% BW | 2–4 sets × 20 s × 2.2 Hz × 115–120% BW | 2–4 sets × 20 s × 3 Hz × 120% BW |
| CMJ | | 5–10 sets × 2–3 reps × 100–105% BW | 5–10 sets × 2–3 reps × 110–115% BW | 5–10 sets × 2–3 reps × 120% BW |
| Hip flexor pull | Hip flexor pull time based criterion for load increment of weekly adjusted 1RM | 8–10 sets × 2–3 reps × 10RM | 8–10 sets × 2–3 reps × 8RM | 8–10 sets × 2–3 reps × 6RM |

**Notes.**

Abbreviations: reps, repetitions; Hz, Hertz; SL, single leg; CMJ, countermovement jump; RM, repetition maximum; BW, body weight.

*de Villarreal et al., 2013*), where differences in total training repetitions between groups was reported to differ by a factor of five.

## Three dimension motion capture (pre and post intervention)

Anatomical markers were placed on the bilateral acromion, manubrium notch, xiphoid process, spinous process of C7 and T10 vertebra, bilateral anterior superior iliac spine, bilateral posterior superior iliac spine, bilateral medial and lateral femoral condyles, bilateral medial and lateral malleoli, 1st and 5th metatarsal heads, superior and inferior tip of posterior calcaneus (*Liew, Morris & Netto, 2016a*). Technical markers were placed as a cluster of four markers on a shell positioned along the lateral aspect of bilateral thigh and shank; as a cluster of two markers on a shell positioned along the lateral surface of the pelvis; and as a single marker on the lateral surface of the calcaneus, bilaterally.

A seven-segment trunk-lower limb kinetic model was created in Visual 3D (C-motion, Germantown, MD) (*Liew, Morris & Netto, 2016a*). Motion and GRF data were captured using an 18 camera system (Vicon T-series, Oxford Metrics, UK) (250 Hz), time synced to in-ground force plates (AMTI, Watertown, MA) (2,000 Hz). Marker trajectories and ground reaction force (GRF) data were low pass filtered at 8 Hz (4th order, zero lag, Butterworth)

**Table 3  Descriptive variables of jump (mean [SD]).**

|  | Isoinertial Pre | Isoinertial Post | Speed-Power Pre | Speed-Power Post |
|---|---|---|---|---|
| CMJ height (cm) | 125.4 (9.6) | 130.2 (9.8) | 128.3 (11.4) | 130.5 (12.3) |
| CMJ depth (cm) | 57.9 (5.7) | 55.5 (4.0) | 55.3 (5.3) | 52.6 (4.0) |
| CMJ maximum knee flexion angle (°) | 107.01 (7.04) | 108.91 (8.04) | 111.29 (7.96) | 114.54 (8.91) |
| SJ height (cm) | 125.0 (8.9) | 128.4 (9.5) | 126.3 (10.4) | 129.6 (12.2) |
| SJ depth (cm) | 59.9 (7.6) | 59.2 (4.6) | 60.0 (5.7) | 57.0 (5.8) |
| SJ maximum knee flexion angle (°) | 98.73 (8.87) | 96.89 (10.95) | 101.13 (10.83) | 98.76 (10.45) |

(*Raffalt, Alkjær & Simonsen, 2016*). All maximal vertical jumps were performed with each foot on a separate force plate, with both arms fixed at 90° abduction to prevent visual obstruction of the anterior torso markers. For both jumps, participants were instructed to descend to a visually estimated 90° knee flexion squat posture. The actual knee flexion angle was quantified post-hoc, but participants were not excluded based on the knee flexion angle. Visual demonstration of the test was provided by the experimenter, practice trials were provided, and when the experimenter was satisfied with the technique, three SJ and three CMJ were performed. Each trial was separated by a 30 s standing rest (*Read & Cisar, 2001*), while each jump type was separated by a minimum of 60 s rest.

## Dependent variables

Descriptive variables of jump mechanics included the lowest center of mass (COM) depth and the knee flexion angle at this posture (Table 3). Dependent variables included: maximal vertical jump height (reached by the COM), power from the hip, knee, and ankle joint. Inverse dynamics was used to calculate joint moments, and joint power was calculated by the dot product of the three-dimensional joint moment with its respective joint angular velocity. In addition, power exchange between the global foot segment and ground interface (termed as the foot joint) was calculated using the Unified-Deformable foot method within Visual 3D (*Takahashi, Kepple & Stanhope, 2012*). For the CMJ, waveforms between the start of the descending phase and toe-off were extracted. For the SJ, waveforms between the start of the ascending phase and toe-off were extracted. Toe-off was considered to occur when the GRF dropped below 20 N (Visual 3D default). The start of the descending phase was defined as a drop in vertical GRF by 2.5% BW (*Meylan et al., 2011*). The start of the ascending phase occurred when the vertical COM velocity ascended at the zero crossing. Total power was defined by the algebraic sum of all eight power waveforms from both limbs. Leg power was defined by the algebraic sum of all four waveforms within a limb. Power waveforms were time-normalized to 101 points.

Raw kinetic variables were normalized to a participant's percentage body mass (M), standing static leg length (L) at each biomechanical testing session and gravitational constant (g) (power to $\%M.L^{0.5}.g^{1.5}$) (*Pinzone, Schwartz & Baker, 2016*). The group's mean normalizing constant of $1\%M.L^{0.5}.g^{1.5}$ was 1,947.5 W.

## Statistical analysis

For all scalar and waveform dependent variables, significant between group differences in change was defined by a non-zero crossing of the Bayesian 95% credible interval (CrI). A 95% CrI provides a range of values for which there is a 95% probability that the interval contains the true regression coefficient. Simple between-group descriptive frequentist parametric and non-parametric statistics were performed, where appropriate, on demographics, resistance training experience, frequency; running experience, frequency, and distance; and training attendance.

### Scalar variable

A Bayesian linear mixed-model with a subject-specific intercept (*Gelman & Hill, 2006*), was used to analyze jump height. Predictor variables of sex (male vs female), time (pre vs post), group (isoinertial vs speed-power), and time-by-group interaction were included. The Bayesian model requires the specification of the prior probability distributions for the regression coefficients for each predictor (Normal distribution), and the variance parameters of the individual data point and each subject (Gamma distribution). In this study, we used "uninformative" priors (i.e., mean of 0, variance of 1,000), which mimics the scenario where the experimenter had no prior knowledge of the relative efficacy of both training groups on jump height. Although there is prior knowledge of intervention effects in vertical jumps (*Markovic, 2007*; *Perez-Gomez & Calbet, 2013*), using an uninformed prior provides a close analogue to traditional ANOVA-based methods. It also means that the posterior knowledge of intervention effects will be driven primarily by the presently collected data.

The Bayesian model was fitted using Markov chain Monte Carlo (MCMC) method in the open source program JAGS (v 4.2.0) with R packages "R2jags" and "rjags". A burn-in period of 1,000 samples was discarded, and 50,000 samples were drawn for inference. Convergence of the MCMC was assessed visually via trace plots.

### Waveform variables

Bayesian functional analysis was performed using the "bayes_fosr" function from the "refund" package (*Goldsmith & Kitago, 2016*). Fixed effect parameters for sex, time, group, and time-by-group interactions, and non-parametric smooth functions (modelled with B-splines) were estimated using a Gibbs sampler with a burn-in of 1,000 and drawing 10,000 inference samples. This number of inference samples was deemed sufficient based on inspection of the predicted to modelled waveforms. The residual covariance structure was estimated using Bayesian functional principle components. For individual joint powers, side was included as a fixed effect. For leg power, the additional fixed effects included were side (right vs left), and side-by-time interaction.

## RESULTS

All 30 participants completed the training and were available for final endpoint analysis.

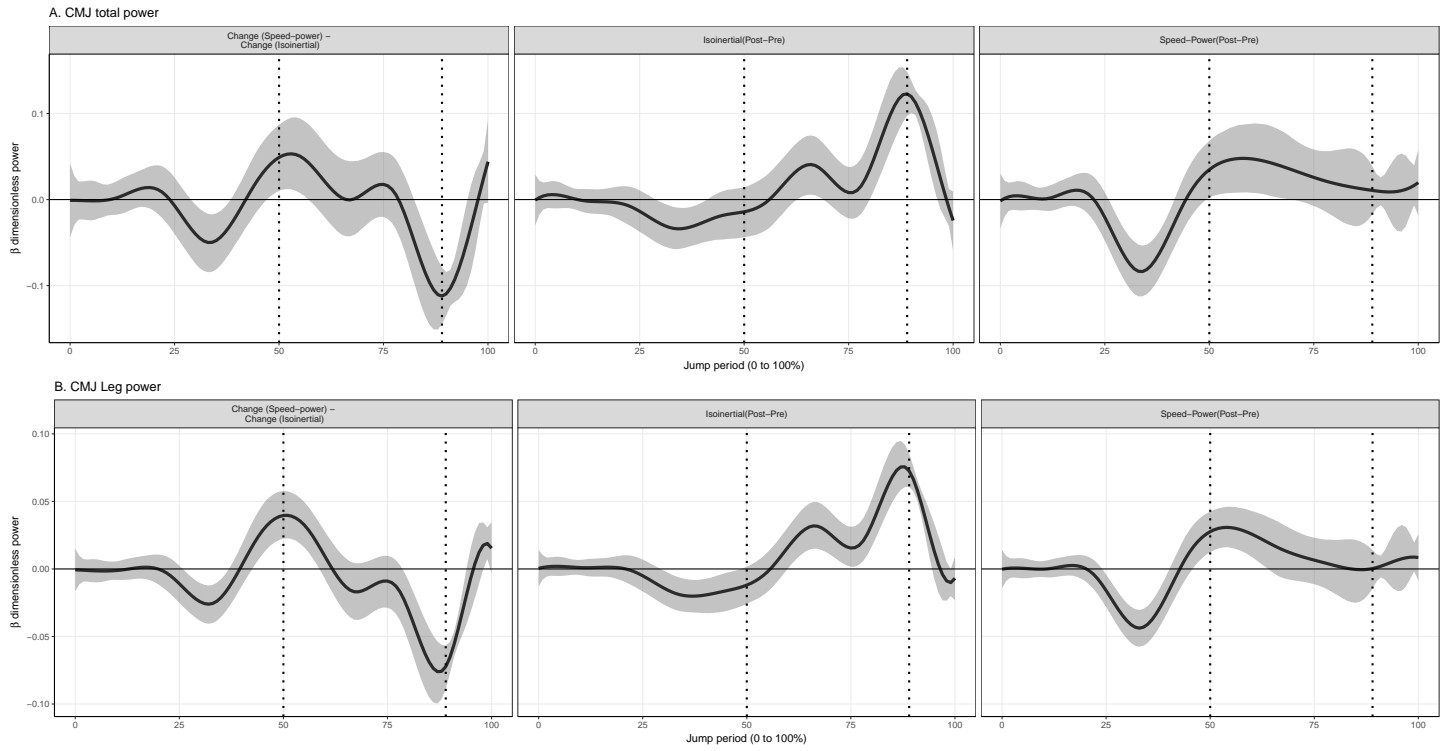

**Figure 2** **Mean and 95% Credible interval of between- and within-group changes to (A) total and (B) leg joint power in countermovement jump.** Vertical dotted lines represents period of the group-averaged peak total power absorption (50% period) and generation (89% period).

### Jump height

Improvement in CMJ height after isoinertial training was 1.95 cm (95% CrI [0.85–3.04] cm) greater than that after speed-power training (a within group change for the isoinertial group of 4.25 cm (95% CrI [3.50–5.00]) cm, compared to a within group change in the speed-power group of 2.30 cm (95% CrI [1.49–3.11] cm)). Improvement in SJ height after isoinertial training was similar to speed-power training with a between group difference in change of −0.23 cm (95% CrI [−1.11–1.60] cm). The within group change in the isoinertial group was 3.34 cm (95% CrI [2.44–4.21] cm), whilst the within group change in the speed-power group was 3.58 cm (95% CrI [2.62–4.55] cm).

### Total joint and leg power ($\%M.L^{0.5}.g^{1.5}$ [95% CrI] (Watts))

Values reported here represent the peak change within the temporal window which was significant. In the descending phase of the CMJ, speed-power training produced a significantly greater increase to total joint and leg power absorption by 0.04% (95% CrI [0.006–0.08]) (80 W) and 0.024% (95% CrI [0.01–0.03]) (49 W), compared to the isoinertial training (Fig. 2). In the ascending phase, isoinertial training produced a significantly greater increase to total and leg power generation by 0.12% (95% CrI [0.08–0.15]) (214 W) and 0.072% (95% CrI [0.06–0.09]) (140 W), compared to the speed-power group (Fig. 2). In the SJ, isoinertial group did not produce a significantly greater increase in total joint and leg power generation compared to the speed-power group (Fig. 3). In both jumps, there
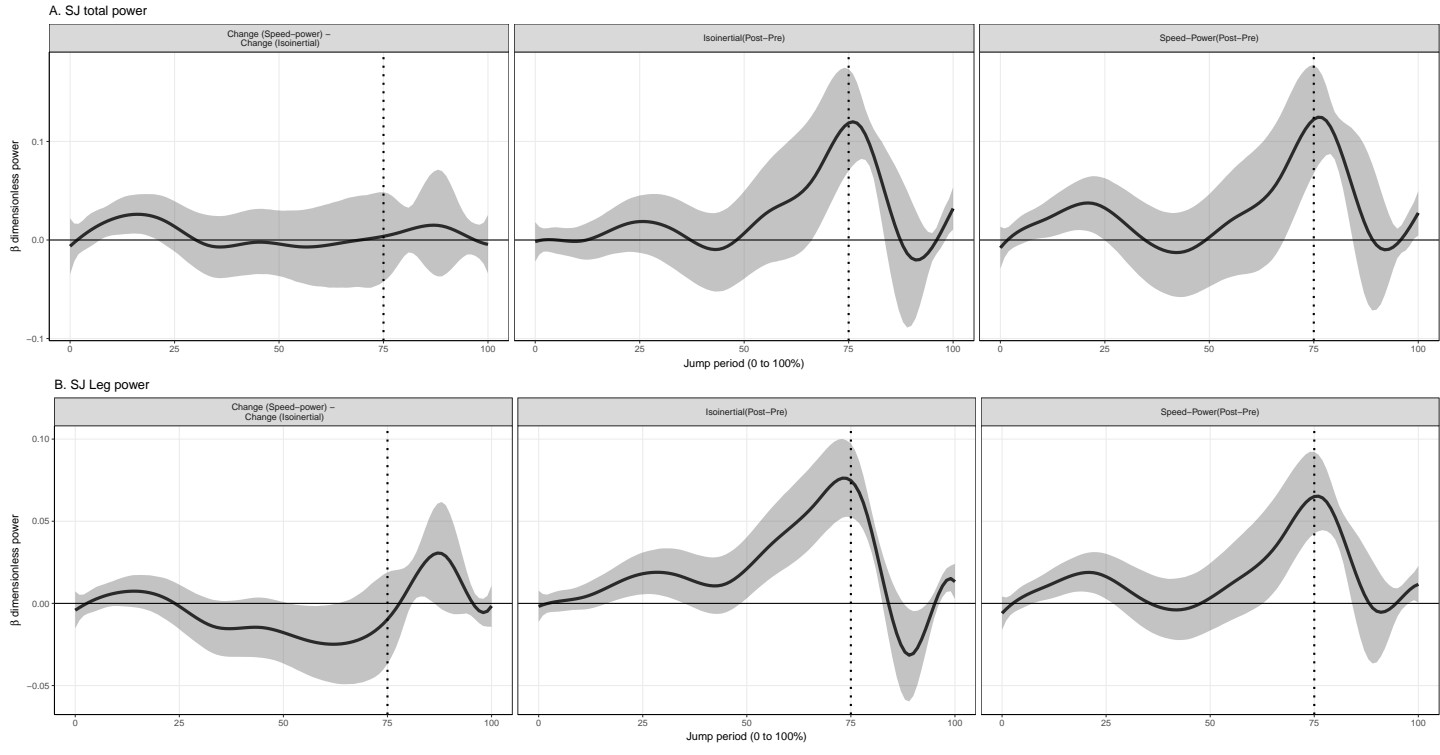

**Figure 3** **Mean and 95% Credible interval of between- and within-group changes to (A) total and (B) leg joint power in squat jump.** Vertical dotted lines represents period of the group-averaged peak total power generation (75% period).

was an absence of a significant side (right vs. left) and side-by-time interaction, indicating that inter-limb power was similar at baseline and post-training.

## Individual joint power (%M.L$^{0.5}$.g$^{1.5}$ [95% CrI] (Watts))

Values reported here represent the peak change within the temporal window which was significant. In the CMJ, isoinertial training produced a significantly greater increase to peak power absorption at the hip by 0.018% (95% CrI [0.007–0.03]) (35 W) and knee by 0.014% (95% CrI [0.006–0.02]) (27 W) during the descending phase, and foot during the ascending phase by 0.011% (95% CrI [0.005–0.02]) (21 W), compared to speed-power training (Fig. 4). Isoinertial training produced a significantly greater increase to power generation at the hip by 0.023% (95% CrI [0.02–0.03]) (45 W), knee by 0.036% (95% CrI [0.02–0.05]) (70 W), ankle by 0.037% (95% CrI [0.02–0.06]) (72W), and foot by 0.019% (95% CrI [0.01–0.03]) (37 W) in the ascending phase, compared to speed-power training (Fig. 4). In the SJ, isoinertial training produced a significantly greater increase in power generation at the ankle by 0.032% (95% CrI [0.009– 0.05]) (62 W), compared to speed-power training (Fig. 5).
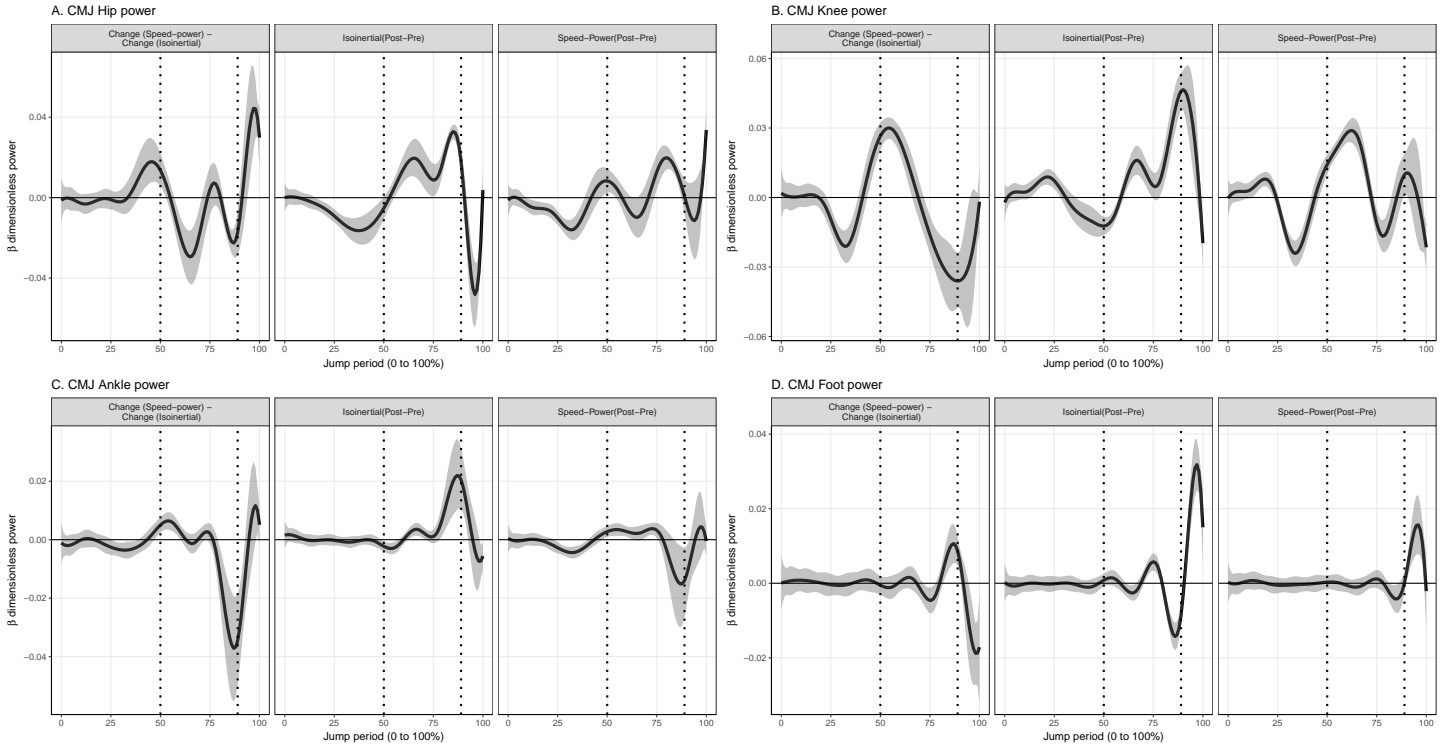

**Figure 4** **Mean and 95% Credible interval of between- and within-group changes to (A) Hip, (B) Knee, (C) Ankle, (D) Foot joint power in coun-termovement jump.** Vertical dotted lines represents period of the group-averaged peak total power absorption (50% period) and generation (89% period).

## DISCUSSION

Although many studies have attempted to elucidate the superiority of different training modalities on vertical jump performance and mechanics, the use of classical ANOVA-based statistics in these studies could have precluded the identification of small, yet potentially beneficial between-intervention differences. Using a Bayesian approach, several major findings were observed in this study involving recreational runners. The findings were: (1) isoinertial training was superior to speed-power training at improving CMJ height; (2) both training approaches significantly improved SJ height, with no significant between group difference; (3) individual joint and inter-joint power changes underlie the superior training effects of isoinertial training on CMJ performance; (4) the principal differential training adaptation between isoinertial and speed-power training for the CMJ was in hip, knee and foot power absorption.

### Bayesian approach to quantify intervention effects

A novel finding of this study was that a significant between-group intervention effect of 2 cm was observed for the CMJ. This effect was opposite to our directional hypothesis, and was not previously reported in previous comparable studies (*Cormie, McGuigan & Newton, 2010a*; *De Villarreal, Izquierdo & Gonzalez-Badillo, 2011*; *Wilson et al., 1993*).
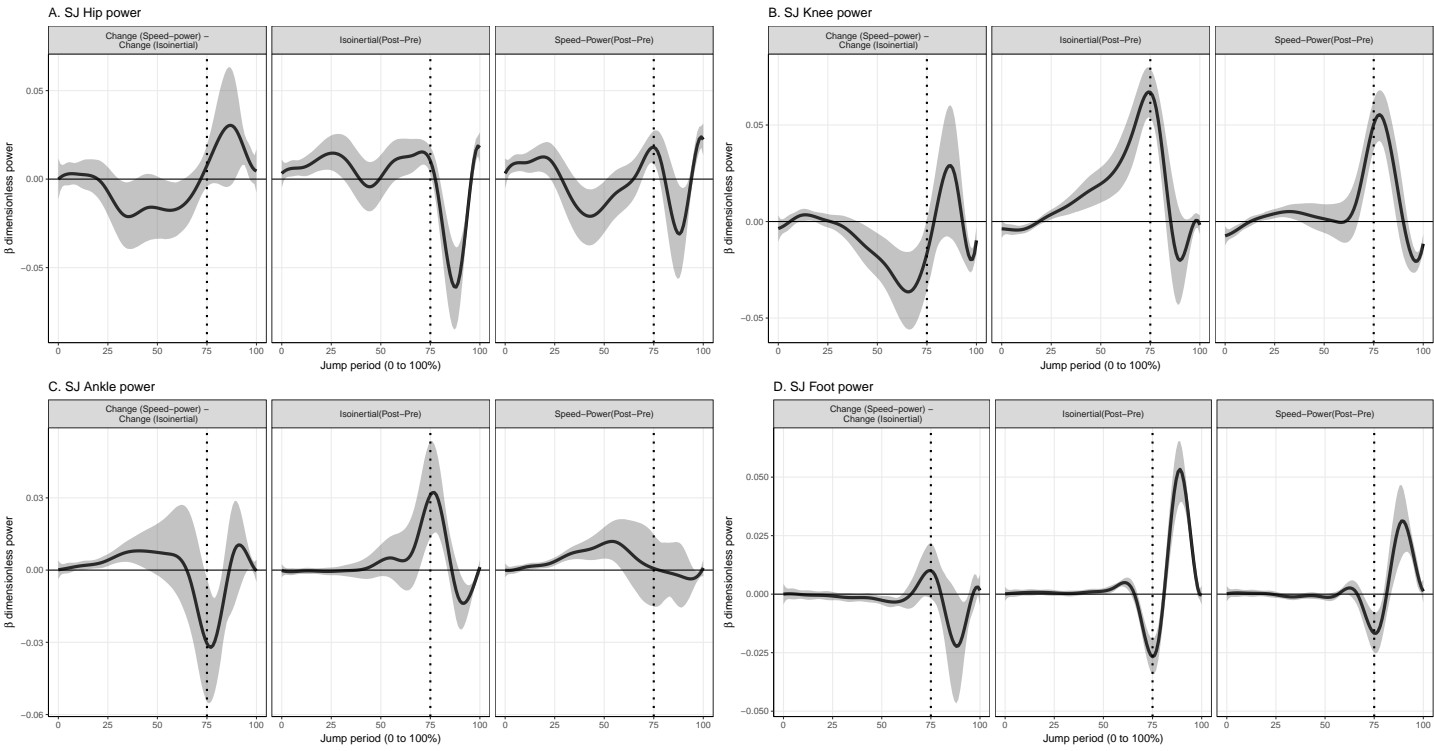

**Figure 5** **Mean and 95% Credible interval of between- and within-group changes to (A) Hip, (B) Knee, (C) Ankle, (D) Foot joint power in squat jump.** Vertical dotted lines represents period of the group-averaged peak total power generation (75% period).

A between-group intervention effect of 2 cm to CMJ performance may be meaningful, given that previous studies have reported that a similar gain in CMJ height was accompanied by improvements to peak running speed in recreational runners by approximately 0.5 km/h (*Taipale et al., 2014*; *Taipale et al., 2013*). It is possible that the between-group changes in CMJ height observed in the present study may be large, or have low variability, such that the choice of Bayesian or Frequentist statistics would have produced similar statistical results. In the present study, the between-group difference in CMJ height gain was approximately 45% of the within-group difference. However, a previous study which reported a maximum within-group change in CMJ of 6 cm, and a between-group difference of 1.9 cm, was still unable to detect a significant between-group change utilizing ANOVA-based inference (*Wilson et al., 1993*).

To the authors present knowledge, the only other study which reported significant between-group differences, reported a greater CMJ height improvement after speed-power compared to isoinertial training (*Newton, Kraemer & Hakkinen, 1999*). It may be that the speed-power training which was previously adopted, used much heavier external loads of 30% to 80% of the 1RM squat load during jump training (*Newton, Kraemer & Hakkinen, 1999*), than used in this study. Alternatively, it may be that isoinertial training more closely matched the participant's underlying muscular deficit, compared to the speed-power training. A greater jump improvement has been observed when matching

the training stimulus to an individual's baseline muscular deficit in producing high force or velocity (*Jimenez-Reyes et al., 2016*). However, this does not completely explain why a between-group difference was observed only in the CMJ and not SJ (see "Joint power mechanisms underlie performance changes" below).

This study determined significance when the 95% CrI did not cross zero, however, it was not the only method of Bayesian inference possible (*Mengersen et al., 2016*). For example, we reported similar intervention effect between the two groups in SJ height gain, when defining significance by a non-zero crossing of the 95% CrI. However, a single Bayesian analysis computes the probabilities across all outcome plausibility (see Figs. S1 and S2). This means that the significance of an intervention effect can be determined based on factors such as: (1) meaningfulness of an effect size, which may be participant-specific, (2) probability of observing the effect, and (3) negative cost of observing the effect. For example, there was a 95% probability that the isoinertial training produced a 1 cm greater CMJ height gain, but only a 12% probability of a 1 cm greater SJ height gain, compared to speed-power training. Low probability to effect size relationships may be meaningful for cohorts where a small change is important.

## Joint power mechanisms underlie performance changes

Three joint-level mechanisms could underlie the superior effect of isoinertial over speed-power training in CMJ height, and not in the SJ. First, greater improvements in hip and knee power absorption in the descending phase of the CMJ after isoinertial training, could underlie the greater improvements to the joints' power generation in ascension, compared to speed-power training. Augmentation of power generation after an eccentric contraction in CMJ have been previously reported, and termed as "eccentric utilization ratio" (*Cormie, McGuigan & Newton, 2010b*; *McGuigan et al., 2006*). Second, greater foot power absorption during peak ankle power generation occurred after isoinertial, and not speed-power training, which could increase the contribution of ankle power to vertical kinetic energy.

The capacity to develop active tension in muscles during the descending phase of the CMJ (termed "preload"), is an important characteristic which distinguishes the CMJ from SJ (*Bobbert & Casius, 2005*). The capacity to generate preload during the descent of a CMJ allows greater power generation to be performed during jump ascent (*Bobbert & Casius, 2005*). This could explain the greater CMJ height gain after isoinertial compared to speed-power training. The increased preload to the knee and hip muscles during the CMJ, was unlikely to have occurred because of an increase in CMJ depth, where the increase in depth was small and within 4 cm in both groups (Table 3). Incidentally, a 4 cm increase in CMJ depth may actually predict a reduced average limb-level power generation by 40 W in a 75 kg individual based on a previously reported regression equation (*Markovic et al., 2014*). This contrasted with an increase in total joint power generation, which may point to a greater non-linear relationship between joint-level and limb-level power measures as CMJ depth increases (*Markovic et al., 2014*).

Rather, greater preload may be due to greater muscle activity after isoinertial training compared to speed-power training, although this theory was not supported by a previous

study which reported that electromyographic activation of the knee vastii muscles did not increase in the descending phase of the CMJ after both speed-power and isoinertial training (*Cormie, McGuigan & Newton, 2010b*). This discrepancy may not be surprising given that joint power represents the net influence of all mono-articular muscles, as well as power transferred from the adjacent joints' muscles (*Prilutsky & Zatsiorsky, 1994*). This means that electromyographic investigations of multiple muscles of the lower limb during vertical jumps, especially that of the hip joint, should be performed in future prospective training studies.

Increased foot power absorption in the ascending phase after isoinertial training could also contribute to the greater CMJ height gain, than speed-power training. When the ankle-foot complex was considered simultaneously, isoinertial training delayed the time to peak ankle-foot power generation, more than speed-power training. This may be beneficial to the energetics of jumping as it allows the segments proximal to the ankle to achieve a more vertical orientation, allowing the ankle-foot power to more effectively contribute to the increase in vertical kinetic energy (*Bobbert & Van Soest, 2001*). Paradoxically, CMJ height has also been reported to improve when foot power absorption was experimentally reduced by wearing stiffer-soled shoes (*Stefanyshyn & Nigg, 2000*). A reduction in foot power absorption would result in an earlier peak in ankle-foot power generation. This necessitates faster hip and knee joint muscles shortening velocities to achieve a vertical orientation of the proximal segments, which may be energetically expensive. This is to ensure that power generation from the ankle-foot contributes effectively to the gain in vertical kinetic energy (*Bobbert & Van Soest, 2001*). Evidently, there are multiple movement strategies to increase jump height, but the body may select the most mechanically efficient strategy in response to training.

The ecological nature of the training precludes knowing the specific parameters responsible for the joint-level changes observed in this study. For example, the isoinertial group had two exercises (leg press and lunge) which have a strong hip and knee extension focus (*Da Silva et al., 2008*; *Riemann et al., 2012*), but the speed-power group only had one (CMJ) (*Raffalt, Alkjær & Simonsen, 2016*). This could have contributed to the greater increase in hip and knee power absorption observed with isoinertial training than speed-power training. A second limitation may be that the inequality of total training volume between groups could also have contributed to the training effects observed. However, between-group differences in training volume has been observed in the literature with equivalent between-group alterations in jump performance (*Cormie, McGuigan & Newton, 2010a*; *De Villarreal, Izquierdo & Gonzalez-Badillo, 2011*). Similar between-group changes in SJ, but not in CMJ, also supports the finding that the observed differences were not simply due to unequal training volume.

It was unlikely that the between-group effect observed in this study was confounded by the exceeding of the prescribed countermovement or starting squat depths of 90° knee flexion in the CMJ and SJ. A greater knee flexion angle during the countermovement phase or starting squat depth has been associated with a greater CMJ and SJ height (*Gheller et al., 2015*). Yet, the isoinertial group which had the smaller increase to CMJ depth of the

two groups, demonstrated the greater increase to CMJ height. In addition, both groups exhibited a reduction in SJ knee flexion angle but an increase in SJ height.

The findings of this study have several broad implications for assessment, training and research methodologies. First, joint-level energetics which include the foot segment, should be assessed in addition to GRF analysis to identify deficient muscle groups requiring additional training as well as the effect of training interventions. Second, any form of training which successfully augments hip, knee, and foot power absorptive capacities, is likely to produce a greater CMJ jump gain, than training which does not alter these mechanics. Third, the present results can go on to form informative priors for future research on vertical jump training practices. Lastly, Bayesian inferences provide a cohesive framework to quantify small but potentially beneficial intervention effects, which may benefit future intervention studies and ultimately sports performance.

## CONCLUSIONS

In conclusion, short-term isoinertial training improved CMJ height more than speed-power training in a group of recreational trained runners with limited resistance training experience. The principle adaptive difference between training modalities was at the level of hip, knee and foot power absorption.

## ACKNOWLEDGEMENTS

The authors of this study would like to thank Nour Faiz Aqil Yaccob, Jason Hu, Nicholas Callaghan, Tess Moynihan, Hannah Watt, and Giorgia Alford for delivering the interventions. The results of this study are presented clearly, honestly, and without fabrication, falsification, or inappropriate data manipulation.

### Funding

Funding for trainers was obtained through a Curtin University internal funding scheme. Bernard Liew was under a postgraduate scholarship "Curtin Strategic International Research Scholarship (CSIRS)". Christopher C. Drovandi was supported by an Australian Research Council's Discovery Early Career Researcher Award funding scheme (DE160100741). The funders had no role in study design, data collection and analysis, decision to publish, or preparation of the manuscript.

### Grant Disclosures

The following grant information was disclosed by the authors:
Curtin University internal funding scheme.
Curtin Strategic International Research Scholarship (CSIRS).
Australian Research Council's Discovery Early Career Researcher Award funding scheme: DE160100741.
## Competing Interests

Justin W.L. Keogh is an Academic Editor for PeerJ.

## Author Contributions

- Bernard X.W. Liew conceived and designed the experiments, performed the experiments, analyzed the data, prepared figures and/or tables, authored or reviewed drafts of the paper, approved the final draft.
- Christopher C. Drovandi and Samuel Clifford analyzed the data, prepared figures and/or tables, authored or reviewed drafts of the paper, approved the final draft.
- Justin W.L. Keogh, Susan Morris and Kevin Netto conceived and designed the experiments, authored or reviewed drafts of the paper, approved the final draft.

## Clinical Trial Ethics

The following information was supplied relating to ethical approvals (i.e., approving body and any reference numbers):

This study was approved by Curtin University's Human Research Ethics Committee (RD-41-14).

## Data Availability

The raw data and code are provided as Supplemental Files.

## Clinical Trial Registration

The following information was supplied regarding Clinical Trial registration:

ACTRN12616000023459.

## Supplemental Information

Supplemental information for this article can be found online at http://dx.doi.org/10.7717/peerj.4620#supplemental-information.

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
