# Peer review of "Joint-level energetics differentiate isoinertial from speed-power resistance training—a Bayesian analysis"

_PeerJ, doi:10.7717/peerj.4620_

## Round 0.1 · original submission · Major Revisions

Thank you for your submission. The manuscript is well written, but requires more detail in several areas to strength the paper. Please address each of the reviewers' comments/questions and provide a response and/or rebuttal to them all. In particular, please address reviewer 1's comments around providing more methodological detail. I look forward to the resubmission.

Scotty

·

Basic reporting

No Comment.

Experimental design

No Comment

Validity of the findings

See General Comments.

Additional comments

The primary aim of this study is in my opinion two-fold and can be summarised as
1) Comparison of 2 different intervention: Isoinertial training vs. speed power training
2) Validity of Bayesian inference to compare between group difference.
The study design is fairly appropriately designed to test for the 2 different interventions. The findings are interesting, the use of Bayesian statistics is novel (and much needed) and the discussions logical.
No objections to publications and in general support its publication. There are a few suggestions below, which I recommend the authors address to strengthen the article.

1) Normalisation of Training (line 107 to 123)
Training programs are logical and sound. It is suggested that the authors include an explanation on how they normalise/equalise the training (e.g. volume etc) between two programs. While I understand this might be hard (especially between the 2 different modalities), doing so will help to convince the readers that the results are more due to the modality and not simply due to the difference in training variables.

2) Leg Stiffness (line 99 to 103)
While, it is understood, that this study is part of a larger study, It is suggested that the authors still include the actual leg stiffness values for 2 main reasons. The first is that the as the authors have mentioned, the study was based (statistically powered) to detect a significant change in leg stiffness (line 99 to 103). Second, while indirect indicators were included (e.g. power absorption, line 205), including the actual leg stiffness values would be useful as it might further reinforce the strength of the results that you shared with the readers.

3) Increase in CMJ depth
It is suggested that the authors try address the increase in CMJ depth a bit further, especially since the difference is one way (greater) and jump power is one of the dependent variables in this study. This is because a greater CMJ depth tends is likely to lead to a greater jump power, though not necessarily jump height (Jidovtseff, Quievre, Harris, & Cronin, 2014). Greater depth can possibly lead to a greater jump power through 1) Greater torque and 2) Increased time for activation and a higher starting force. Since jump power is one of the main dependent variables in this study, it is recommended, that the authors elaborate this slightly further.

On a side note, I concur with the discussion point that the increased preload might explain the greater CMJ height is logical.

·

Basic reporting

1. L 32: Please provide a reference for the statement: “squat jumps and countermovement jumps are the most commonly assessed vertical jump types”
2. L 46: Paragraphs should not start with the word “yet”
3. L 49-51: The authors cite several previous studies that have compared interventions. They decided that another study utilising Bayesian methods was the best way to identify the small effect sizes previously observed. Would a meta-analysis of these studies have been a better way to pick them up?
4. L 52: “have” should be “has”
5. L 56: Would “qualitative” be better described as “descriptive”?
6. L 59 (and elsewhere): “clinical” – should this word be deleted? The decisions aren’t necessarily in a clinical setting
7. L 75: Consider adding “if any” after “which”
8. L 81: Clarify that you hypothesise increases in both CMJ and SJ jump height.
9. L 81: Please define an improvement in energetics
10. L 200: The data in brackets reads as SJ height change not the diff between isoinertial change vs speed-power change. Please clarify
11. L 254: Consider deleting the word “presently”
12. L 255: “was” should be “has been”
13. L 259-60: Consider changing “which does not mean it was the only method of Bayesian inference” to “however it was not the only method of Bayesian inference used”
14. L 267: Insert “the” after “that”
15. L 288: Consider changing “This was” to “Although this theory is”
16. L 291: Add a full stop after “2010b)”
17. L 311: Consider changing “active ingredients” to “specific parameters” (no quotes needed)
18. L 335: Change “ultimate” to “ultimately”
19. Figure 1: Correct spelling of “enrollment” to “enrolment”
20. Figure 1: Expand or otherwise clarify “mocap”
21. Figure 2 & 3: Provide more detail on the calculation of the location of the dotted lines – e.g. is this from both groups combined?
22. Figure 2 & 3: 18 graphs on one page was too many for easy evaluation. Consider superimposing to allow easier comparisons. Some graphs could maybe go into the supplementary material. Shading was also too light when printed out (black/white). The mean line was not visible.

Experimental design

1. The sample size calculation is for an outcome (leg stiffness) not reported in the current study (as this analysis was part of a larger study) and therefore it’s utility is not clear. While I am unsure if there are validated power calculations for Bayesian analyses; it would inform the reader if the authors could state the sample size that would have been needed if jump height was considered the primary outcome (using the minimum meaningful difference that they hoped to be able to detect / expected from previous studies).

2. L 120: How does the volume, intensity, repetitions and time to complete the CMJ and SL exercises compare to the resistance training? Could your results be due to differences in these rather than the modality?

3. This study utilises a similar intervention methodology to other papers; but differs by the type of analysis (Bayesian vs frequentist). The introduction implies that a Bayesian analysis was chosen a priori however it is not clear why a larger sample size was not considered given the small effect sizes found in previous studies.

4. Would a comparison to the frequentist methodology be more appropriate to demonstrate superiority of the Bayesian method?

5. L 17 & 167 & Results: While I am not overly familiar with the use of Bayesian methods (as I expect many readers will be as well), it surprised me that a level of significance was defined in a similar way to frequentist methods. From my understanding Bayesian methods aim to get away from the binary classification of results so that small effects are not dismissed but given a level/range of confidence/credibility.

6. A copy of the visual 3D model file and analysis pipelines would improve reproducibility

7. L 153: Please explain / reference the unified-deformable foot method

8. L154-5: Please define/reword your use of the terms eccentric and concentric as muscle action was not measured directly

9. L 183-4 & 190: How are the number of burn-in and inference samples decided? Why do they differ between scalar and waveform variables?

10. L 192: The correct analysis of left/right sides has been debated (Menz 2004, Stewart 2018 - http://www.gaitposture.com/article/S0966-6362(17)30974-8/fulltext ). How does your analysis compare to the Stewart approach (other than being Bayesian)?

11. Table 1: p-values are described (frequentist analysis) but the tests used are not in the methods. Why use frequentist for some analyses and not others? Given the use of males and females, were body mass and height data binomially distributed? Were these analysed differently?

12. Table 1: Resistance training experience is reported but no mention of speed-power training experience (e.g. from circuit training).

13. Table 1: Does the lower resistance training experience and recent training in the isoinertial group explain the greater increases observed?

14. Table 2: Is the total time to carry out each session the same for isoinertial vs speed-power?

Validity of the findings

1. L 22: no variability is reported for the increase in power absorption. Is the value reported a mean?
2. L 205-208: Please include CrIs. Are these mean values?
3. L 225: “meaningful” – please define in the introduction and/or in relation to the sample size calculation
4. L 238: “average” – is this a mean?
5. L 242: “may be large” – consider adding “or have low variability”
6. L 247: Did the Wilson study have a similar sample size?
7. L 253: “may be plausible” - Can you be more definitive than this?
8. L 286: “qualitatively similar” – this is measured in quantities – consider stating instead that the means were within 5 degrees.
9. L 319: I presume these reports used frequentist methodology – were the effect sizes similar directions and magnitudes but just not detected?
10. Table 3: Please include info on CMJ and SJ height

Additional comments

No further comments.

---

## Round 0.2 · accepted · Accept

Congratulations on acceptance. I feel that you provided sufficient rationale for your rebuttal and have addressed both reviewers' comments adequately. Best of luck with your future work!
Scotty